# Glissonean Pedicle Isolation Focusing on the Laennec’s Capsule for Minimally Invasive Anatomical Liver Resection

**DOI:** 10.3390/jpm13071154

**Published:** 2023-07-18

**Authors:** Mamoru Morimoto, Yoichi Matsuo, Keisuke Nonoyama, Yuki Denda, Hiromichi Murase, Tomokatsu Kato, Hiroyuki Imafuji, Kenta Saito, Shuji Takiguchi

**Affiliations:** Department of Gastroenterological Surgery, Nagoya City University Graduate School of Medical Sciences, Kawasumi 1, Mizuho-Cho, Mizuhoku, Nagoya 467-8601, Japan; mamoru11@med.nagoya-cu.ac.jp (M.M.); knonoyama0924@gmail.com (K.N.); woody2shoes06@gmail.com (Y.D.); muramen5.com@gmail.com (H.M.); tomo.k.g.w@gmail.com (T.K.); himafuji@med.nagoya-cu.ac.jp (H.I.); ksaito@med.nagoya-cu.ac.jp (K.S.); takiguch@med.nagoya-cu.ac.jp (S.T.)

**Keywords:** anatomical liver resection, Glissonean pedicle isolation, Laennec’s capsule, laparoscopic surgery, robotic surgery

## Abstract

**Background:** Inflow control is one of the most important procedures during anatomical liver resection (ALR), and Glissonean pedicle isolation (GPI) is one of the most efficacious methods used in laparoscopic anatomical liver resection (LALR). Recognition of the Laennec’s capsule covering the liver parenchyma is essential for safe and precise GPI. The purpose of this study was to verify identification of the Laennec’s capsule, to confirm the validity of GPI in minimally invasive surgery, and to demonstrate the value of GPI focusing on the Laennec’s capsule using a robotic system that has been developed in recent years. **Methods:** We used a cadaveric model to simulate the Glissonean pedicle and the surrounding liver parenchyma for pathologic verification of the layers. We performed 60 LALRs and 39 robotic anatomical liver resections (RALRs) using an extrahepatic Glissonean approach, from April 2020 to April 2023, and verified the layers of the specimens removed during LALR and RALR based on pathologic examination. In addition, the surgical outcomes of LALR and RALR were compared. **Results:** Histologic examination facilitated by Elastica van Gieson staining revealed the presence of Laennec’s capsule covering the liver parenchyma in a cadaveric model. Similar findings were obtained following LALR and RALR, thus confirming that the gap between the Glissonean pedicle and the Laennec’s capsule can be dissected without injury to the parenchyma. The mean GPI time was 32.9 and 27.2 min in LALR and RALR, respectively. The mean blood loss was 289.7 and 131.6 mL in LALR and RALR, respectively. There was no significant difference in the incidence of Clavien–Dindo grade ≥III complications between the two groups. **Conclusions:** Laennec’s capsule is the most important anatomical landmark in performing a safe and successful extrahepatic GPI. Based on this concept, it is possible for LALR and RALR to develop GPI focusing on the Laennec’s capsule. Furthermore, a robotic system has the potential to increase the safety and decrease the difficulty of this challenging procedure.

## 1. Introduction

The clinical value of an anatomical hepatectomy for hepatocellular carcinoma has recently been reported [1]. Of note, a non-anatomical, tissue-sparing hepatectomy is associated with worse disease-free survival in patients with KRAS-mutated tumors. Therefore, a more extensive anatomical hepatectomy for KRAS-mutated tumors may be warranted [2]. Precise inflow control is of utmost importance for precise anatomical liver resection (ALR). There are several methods of inflow control; however, a consensus has emerged that Glissonean pedicle isolation (GPI) is the safest and most reliable method in the case of laparoscopic anatomical liver resection (LALR) [3]. In addition, the Laennec’s capsule has been noted as an important anatomical landmark for precisely and safely performing a GPI [4]. The Glissonean pedicle is surrounded by connective tissue (Walaeus sheath), which encompasses the portal triad. More precisely, the Walaeus sheath is a separate entity from the peritoneum and the liver capsule (Laennec’s capsule). For this reason, it is important to understand that the Laennec’s capsule can be separated from the Walaeus sheath at the hepatic hilus and that Glissonean pedicles wrapped by the Walaeus sheath can be safely approached extrahepatically [5].

Most of the previous reports involving Laennec’s capsule have verified the existence of the Laennec’s capsule, but it is not known whether the Laennec’s capsule is a useful landmark during surgery. In addition, motion restrictions complicate laparoscopic GPI, which is currently performed in only a few specialized centers. In contrast, the highly flexible manipulation and precise and stable field of view afforded by a robotic system reduce the difficulty of GPI and increase the safety. Therefore, we believe that a robotic system can perform more precise procedures than conventional laparoscopic surgery; however, to date, there have been no reports comparing the results of GPI focusing on the Laennec’s capsule in ALR with those of laparoscopic and robotic surgeries, and the usefulness of robotic surgery has not been established.

Therefore, we performed a pathologic examination of the Glissonean pedicle and the surrounding layers using a cadaver model to delineate the Laennec’s capsule. Furthermore, we attempted to verify the usefulness of a robotic system in ALR based on a comparison of the LALR and RALR outcomes with GPI focusing on the Laennec’s capsule.

## 2. Material and Methods

### 2.1. Cadaver Simulation

Three Thiel-embalmed cadavers were used for this study, which was conducted between April and September 2016. The cadavers were donated to the Department of Anatomy (Nagoya City University). The patients had signed documents agreeing to donate their body for use in clinical studies. The format of the document met the criteria for the Japanese “Act on Body Donation for Medical and Dental Education”. All cadavers were embalmed using the method described by Thiel [6,7,8]. The cadavers were embalmed in a water-based solution consisting of salt with a small amount of formaldehyde for fixation, boric acid for disinfection, glycol, chlorocresol, and ethanol; this precipitation results in tissue homogenization. The skin was life-like, and the joints were fully flexible. To confirm the existence of the Laennec’s capsule after a laparoscopic extrahepatic GPI, the separated Glissonean pedicle and surrounding liver parenchyma specimens in the cadaveric model were stained with Elastica van Gieson and evaluated.

### 2.2. Extrahepatic Glissonean Pedicle Isolation in LALR and RALR

Only cases in which an extrahepatic GPI was used as an inflow control in LALR and RALR were included in the study. There are three types of GPI during ALR. The Glissonean approach can be defined as extrahepatic with no parenchymal transection or as intrahepatic when minimal parenchymal dissection or additional parenchymal transection is necessary to access the Glissonean pedicle [3]. To preserve the Laennec’s capsule, the extrahepatic Glissonean approach must be performed, which was performed according to the Gate theory, as proposed by Sugioka [4]. 

We used scissors with a small, thin tip during LALR to obtain fine vision in a narrow and limited field of view without any obstruction from the instruments. The da Vinci Surgical Xi system^®^ (Intuitive Surgical, Sunnyvale, CA, USA) was used in RALR. Maryland bipolar forceps^®^ (Intuitive Surgical, Sunnyvale, CA, USA) were used in all cases as the surgeon’s right hand during GPI and liver parenchymal transection. Fenestrated bipolar forceps^®^ (Intuitive Surgical, Sunnyvale, CA, USA) were used as the surgeon’s left hand and Cadiere forceps^®^ (Intuitive Surgical, Sunnyvale, CA, USA) were used for liver traction. All cases included in this study involved GPI using Maryland bipolar forceps^®^. The Pringle maneuver was used in all laparoscopic and robotic procedures.

### 2.3. Patient Characteristics

Patient background data included age, gender, body mass index, histologically proven cirrhosis (postoperative evaluation), indocyanine green (ICG) retention rate at 15 min (ICGR15), indication for liver resection, and previous liver resection. The terminology for liver anatomy and hepatectomy procedures was primarily based on the Brisbane 2000 Terminology of Liver Anatomy and Resections [9]. Anatomical resection (AR) was classified into left lateral sectionectomy (LLS), segmentectomy, sectionectomy, and hepatectomy resecting three or more segments. 

### 2.4. Peri- and Post-Operative Outcomes

Intraoperative outcomes were evaluated based on total operative time, duration of the extrahepatic GPI, measured blood loss, and operative difficulty according to the IWATE criteria [10]. Postoperative outcomes were evaluated by complications graded according to the Clavien–Dindo (C-D) classification [11] and the length of postoperative hospital stay. Overall and major complications were defined as those complications occurring within 90 days of any C-D complications and ≥grade IIIa, respectively. 

### 2.5. Statistical Analysis

All measurement data are expressed as a mean ± standard deviation (SD). The Mann–Whitney U test was used to compare continuous variables between two independent groups. As appropriate, categorical variables were analyzed using Fisher’s exact test or a chi-squared test. Significance was defined at a *p*-value <0.05. All statistical analyses were performed using EZR (Saitama Medical Center, Jichi Medical University (http://www.jichi.ac.jp/saitama-sct/SaitamaHP.files/statmedEN.html (accessed on 25 May 2023)), Kanda, 2012), a graphical user interface for R (version 2.13.0, The R Foundation for Statistical Computing, Vienna, Austria). More precisely, EZR is a modified version of R Commander (version 1.6-3) that was designed to add statistical functions frequently used in biostatistics. 

## 3. Results

### 3.1. Surgical Procedure

#### 3.1.1. The Laparoscopic Extrahepatic Glissonean Approach and Pathologic Examination of the Laennec’s Capsule in Cadaveric Models (Appendix A)

The Laennec’s capsule covers not only the entire surface of the liver parenchyma beneath the serosa, including the bare area, but also the intrahepatic parenchyma surrounding the Glissonean pedicles and the plate system. At first glance, the serosa and the Laennec’s capsule appear to be the same membrane; however, the serosa and the Laennec’s capsule are distinct structures. Indeed, after the serosa is divided, a clear boundary between these membranes can be confirmed. First, we divide the serosa covering the left Glissonean pedicle and the liver parenchyma in a cadaveric model. The right aspect of the left Glissonean pedicle is exposed while preserving the Laennec’s capsule to the liver parenchyma (Figure 1A). Then, the dissection can be performed with a direct view of the dorsal surface of the left Glissonean pedicle. The liver parenchyma in the cadaver is much more fragile than the liver parenchyma in vivo and therefore, is easily injured. Preserving the Laennec’s capsule, however, can prevent injury to the parenchyma. Preservation of the Laennec’s capsule also allows for safe dissection of the thin, dorsally branching Glissonean pedicles that are difficult to access (Figure 1B). Then, the left lateral segment is mobilized and the arantius plate and the liver parenchyma are detached along the Laennec’s capsule. The dorsal surface of the left Glissonean pedicle is separated within the same layer, which can be connected to the previously detached layer. We attempted to elucidate the layer structure around the Glissonean pedicle histologically using Elastica van Gieson staining. We confirmed the existence of a membrane around the liver parenchyma in cadaveric models. A dense fibrous layer was observed around the liver parenchyma in the left GPI (low-power field (Figure 1C) and high-power field (black arrows, Figure 1D).

#### 3.1.2. The Laparoscopic Extrahepatic Glissonean Approach and Pathologic Examination of the Laennec’s Capsule in a Live Body


**①**
**Laparoscopic extrahepatic right anterior and posterior GPI (Appendix A)**


Isolation of the right anterior and posterior Glissonean pedicle and second-order branches of the Glissonean pedicle were performed during right hepatectomy in a live body. First, a cystic plate cholecystectomy was performed to expose the Laennec’s capsule (Figure 2A). Following the cystic plate cholecystectomy, the same layer of dissection was continued to the ventral aspect of the right anterior Glissonean pedicle to expose the Laennec’s capsule (Figure 2B). Thus, the laparoscopic surgery magnifying effect allowed the presence of the Laennec’s capsule to be continuously recognized. 


**②**
**Laparoscopic extrahepatic Glissonean pedicle 8 isolation (Appendix A)**


Isolation of the Glissonean pedicle 8 and the third-order branches of the Glissonean pedicle during segmentectomy 8 in a live body was performed. First, the boundary between the cystic plate and the Laennec’s capsule covering the liver parenchyma was identified, and dissection was continued in this same layer from the root of the right anterior Glissonean pedicle toward the tertiary branch (Figure 2E). Several thin Glissonean pedicle branches extend from the Glissonean pedicle 8, but continued dissection in the correct layer minimized the possibility of unintentional damage. Blunt dissection was performed to isolate the Glissonean pedicle 8 while ensuring the dorsal space (Figure 2F). A specimen from the right anterior Glissonean pedicle and Glissonean pedicle 8 with the surrounding liver parenchymal tissue attached was prepared, which then underwent a pathologic examination. A dense fibrous layer was observed on the liver parenchyma in the right anterior GPI in a low-power field (Figure 2C), in a high-power field (black arrow, Figure 2D), in Glissonean pedicle 8 isolation in a low-power field (Figure 2G), and in a high-power field (black arrow, Figure 2H). Indeed, these membranes were the Laennec’s capsule, thus confirming that the gap between the Glissonean pedicle and the Laennec’s capsule could be explored without parenchymal injury.

#### 3.1.3. The Robotic Extrahepatic Glissonean Approach and Pathologic Examination of the Laennec’s Capsule in a Live Body


**①**
**Robotic extrahepatic right posterior Glissonean pedicle isolation (Appendix A)**


Isolation of the right posterior Glissonean pedicle and the second-order branches of the Glissonean pedicle during posterior sectionectomy of a live body was performed using a robotic system. Clear recognition of the Laennec’s capsule by taking advantage of the laparoscopic surgery magnification effect and relaxation of movement restrictions by the highly flexible movements of a robotic system may allow for a safer and more precise GPI. Although the junction of the right posterior and anterior Glissonean pedicles is a very tight space, the Laennec’s capsule can be clearly exposed by a very delicate manipulation that takes advantage of the characteristics of the robotic system (Figure 3A). The Laennec’s capsule on the dorsal aspect of the right posterior Glissonean pedicle can be exposed without destroying the liver parenchyma (Figure 3B).


**②**
**Robotic extrahepatic Glissonean pedicle 4a and 4b isolation (Appendix A)**


An extrahepatic Glissonean approach was performed to the third-order branches. In this case, Glissonean pedicles 4a, 4b, and 4c were pre-isolated, and blood flow was occluded to identify the territory of segments S4a, S4b, and S4c. Dissection of the round ligament and pulling the round ligament caudally facilitated easy entry between the umbilical plate and the Laennec’s capsule (Figure 3E). If the dissection is performed focusing on the Laennec’s capsule, all branches can be isolated without misidentification, while directly observing the branches (Figure 3F). Finally, Glissonean pedicles 4a, 4b, and 4c were all successfully isolated using the extrahepatic approach. 

Further, a pathologic examination proved that GPI was performed with the Laennec’s capsule completely preserved on the liver parenchyma. A dense fibrous layer was observed on the liver parenchyma, in the right posterior GPI in a low-power field (Figure 3C), in a high-power field (black arrow, Figure 3D), in Glissonean pedicle 4b isolation in a low-power field (Figure 3G), and in a high-power field (black arrow) (Figure 3H).

### 3.2. Patient Characteristics (Table 1)

This study included patients who underwent surgical resection of solitary liver tumors based on preoperative radiologic images from April 2020 to April 2023 after cadaver simulation. A total of 99 patients underwent LALRs and RALRs with extrahepatic GPI at Nagoya City University. All operations were performed by one experienced laparoscopic surgeon (M.M.) with endoscopic surgeon qualifications from the Japan Society for Endoscopic Surgery. The indication for use of a minimally invasive technique followed practical guidelines based on the second international laparoscopic liver consensus conference [12]. Informed consent was obtained from each patient prior to surgery.

The patient characteristics are listed in Table 1. We performed LALRs and RALRs in 60 (61%) and 39 patients (39%), respectively. Between the LALR and RALR groups, there were no significant differences with respect to mean age and gender distribution (mean age, 69.3 ± 10.5 years in the LALR group vs. 69.6 ± 9.5 years in the RALR group; male/female ratio, 34:26 in the LALR group vs. 23:16 in the RALR group). In addition, the mean body mass index was similar in the two groups. The prevalence of liver cirrhosis was 25% and 23% in the LALR and RALR groups, respectively; this difference was not statistically significant. The ICGR15 was similar in the two groups. Furthermore, the indications for liver resection did not differ significantly between the two groups. Eight patients (13%) in the LALR group and five patients (13%) in the RALR group had undergone a previous liver resection.

### 3.3. Types of Liver Resection (Table 2)

Table 2 shows the types of liver resection; there was no significant difference in the procedure between the LALR and RALR groups. A right hepatectomy was performed in eight LALR cases and two RALR cases, while a left hepatectomy was performed in twelve LALR cases and five RALR cases. There were twenty sectionectomies: ten left lateral sectionectomies (five LALR cases and five RALR cases); three left medial sectionectomies (one LALR case and two RALR cases); two right anterior sectionectomies (all LALR cases); and five right posterior sectionectomies (four LALR cases and one RALR case). There were forty-six segmentectomies: segment I in five cases (three LALR cases and two RALR cases); segment II in five cases (three LALR cases and two RALR cases); segment III in four cases (one LALR cases and three RALR cases); segment IVa in two cases (one LALR case and one RALR case), segment IVb in six cases (two LALR cases and four RALR cases); segment V in five cases (two LALR cases and three RALR cases); segment VI in four cases (all LALR cases); segment VII in seven cases (all LALR cases); and segment VIII in eight cases (five LALR cases and three RALR cases). 

### 3.4. Operative Data (Table 3)

The operative data are presented in Table 3. The mean operative time was 399.8 ± 109.5 and 366.1 ± 98.3 min for LALR and RALR, respectively; there was no significant difference between the two groups. The mean duration of GPI was 32.9 ± 11.6 and 27.2 ± 9.3 min for LALR and RALR, respectively (*p* < 0.05). The mean blood loss was significantly lower in the RALR group (131.6 ± 102.2 mL) than the LALR group (289.7 ± 303.8). The mean postoperative hospital stay was similar between the two groups (12.8 ± 8.5 days (LALR) and 11.3 ± 3.9 days (RALR)). The complication rate (Clavien–Dindo classification, Grade ≥ IIIa) was 3.3% in the LALR group and 5.1% in the RALR group (no statistically significant difference).The complications included two cases of abdominal abscess in the LALR group, and one bile leak and one abdominal abscess in the RALR group. There were no postoperative mortalities in the two groups.

## 4. Discussion

The first objective of this study was to verify the presence of the Laennec’s capsule. The second objective of was to determine the impact of laparoscopic and robotic surgery on GPI with an emphasis on the Laennec’s capsule serving as an anatomical landmark. Using cadavers and live bodies, we were able to demonstrate the presence of the Laennec’s capsule covering the liver parenchyma. Then, we compared the operative outcomes of LALR and RALR with extrahepatic GPI and showed that the inflow control time and blood loss were significantly shorter and less, respectively, with robotic surgery. Furthermore, there was no difference in the incidence of postoperative complications, suggesting that robotic surgery may reduce procedure challenges and improve safety compared to laparoscopic surgery.

The benefits of minimally invasive surgery for liver tumors have been clearly demonstrated in the last two decades [13]. The number of laparoscopic liver resection procedures has increased markedly, with favorable results, such as fewer complications and blood transfusions, less blood loss, and shorter hospital stays compared to those with open surgery as well as similar oncologic outcomes [14,15]. ALR has been reported to be a safe and useful procedure for hepatocellular carcinoma, and thus, standardization of laparoscopic ALR is urgently needed [16,17]; however, laparoscopic anatomical hepatectomy is a difficult surgical procedure that can only be performed in a few specialized centers. ALR is defined as the complete removal of the liver parenchyma confined within the responsible portal territory [18]. Therefore, there is a consensus that accurate inflow control is most important for accurate ALR, and that GPI is the most reasonable method for laparoscopic resection.

The Glissonean approach can be defined as extrahepatic with no parenchymal transection or as intrahepatic when minimal parenchymal dissection or additional parenchymal transection are necessary to access the Glissonean pedicle [3]. We believe that the extrahepatic GPI approach is the most precise and safe inflow control method. The intrahepatic Glissonean approach is also a potentially effective method, but it may lead to misidentification of the Glissonean pedicles and unnecessary injury to the small branches of the Glissonean pedicles. During an intrahepatic Glissonean approach, the hepatic vein is often used as a landmark and the liver parenchyma is dissected first; however, the hepatic vein is not as accurate as identifying the segment with prior portal vein occlusion using the extrahepatic Glissonean approach. Indeed, the extrahepatic Glissonean approach is the most accurate option for performing an ALR, and prior isolation of the responsible Glissonean pedicle is essential. Sugioka et al. [4] reported the importance of the Laennec’s capsule as an anatomical landmark when performing an extrahepatic GPI; however, there have been few reports on the usefulness of GPI focusing on the Laennec’s capsule [19,20]. When performing an extrahepatic GPI, without a clear understanding of the layer structure around the Glissonean pedicle, the recognition of the boundaries is unclear and may enter into the Glissonean sheath and cause damage to the artery, portal vein, and bile ducts.

In 1802, Laennec [21] first described a membrane as a distinct structure from the serosa. Couinaud [22] established the concept of the plate system as a fibrous thickening part of the Glissonean sheath and demonstrated that the Laennec’s capsule had no continuity with the Glissonean pedicle [23]. Hayashi et al. [24] conducted a precise histologic study of cadaveric livers with elastic fiber and lymphatic vein staining and revealed that the so-called Glissonean capsule was not derived from the Glissonean sheath, but from the Laennec’s capsule surrounding the pedicles, and that the Laennec’s capsule extended to the peripheral Glissonean pedicles [24]. Theoretically, the Laennec’s capsule should be surrounding all the Glissonean pedicles, but most reports, to date, have focused on the irst-order branches. In the case of sectionectomy and segmentectomy, second- and third-order branches need to be isolated, but the surrounding layer has not been explored. An extrahepatic GPI of the first-order branches focusing on the Laennec’s capsule is not a very challenging procedure because it can be performed under good vision. Recognition of the Laennec’s capsule in second- and third-order branches is more important than detection of the Laennec’s capsule in first-order branch isolation. In vivo pathologic examination allowed us to verify the presence of the Laennec’s capsule around first-, second-, and third-order branches; however, because the Laennec’s capsule around the Glissonean pedicle is very thin and fragile, the Laennec’s capsule is easily disrupted by an intrahepatic GPI, which makes isolation of the Glissonean pedicle at the appropriate layer very difficult. Although the extrahepatic third-order branches are more challenging due to their anatomical location, a precise and safe GPI can be performed by following the Laennec’s capsule from the hilum as a landmark. In principle, the extrahepatic approach is considered reasonable for GPI.

Robotic surgery has gained growing acceptance in recent years, including liver resection [25,26]. The robotic approach, with its added degrees of freedom, improved visualization, stability of the robotic platform, and better ergonomics, improves surgeon dexterity during complex minimally invasive procedures. There is a recent interest in robotic liver surgery and the number and complexity of procedures are rapidly increasing [27,28,29]; however, the benefit of robotic surgery for ALR is still under debate with limited evidence available. We believe that the advantages of the robotic system can be utilized to perform ALR more precisely and safely than open or laparoscopic surgeries. In addition to the conventional laparoscopic surgery magnification effect, the robotic system provides a stable field of view with 3D high-resolution images without tremor. The stable field of view makes recognition of the Laennec’s capsule easier than in laparoscopic surgery. In addition, stabilization of the articulated function and the motion scale of robotic forceps allow for more delicate manipulation. It is essential to perform very delicate manipulations to expose the very thin and fragile Laennec’s capsule. Based on the above, the advantages of a robotic system and GPI focusing on the Laennec’s capsule is so compatible that RALR has been actively introduced since 2020 in our institute.

We reviewed the surgical outcomes of LALR and RALR. All patients underwent an extrahepatic Glissonean pedicle isolation, with no significant differences in patient background. According to the level of difficulty, the average scores were 7.7 and 7.4 for LALR and RALR in advanced levels, respectively. Re-hepatic resections were required for 13% of LALRs and RALRs, and are expected to be even more difficult due to adhesions and other factors. With respect to surgical classification, various types of extrahepatic GPI were performed in full, ranging from first-, second-, and third-order branches of the Glissonean pedicles. Comparison of surgical outcomes showed a shorter GPI time and lower blood loss during RALR. The postoperative complication rates did not differ between the two groups, and the safety of RALR was shown to be equivalent to LALR. Complications in the LALRs were intra-abdominal abscess without bile leak in two cases; complications in the RALRs were bile leak and subcutaneous abscess without bile leak. The bile leak in the RALR was unlikely to have been caused by extrahepatic GPI because the bile leak was from the peripheral bile duct and there was no injury to the hilar bile duct. A cavitron ultrasonic surgical aspirator (CUSA), which is widely used for liver parenchymal resection in LALR, cannot be attached to a robotic system. A disadvantage of the robotic system is that it does not have a specifically designed parenchymal resection instrument, which may have prevented the widespread use of the robotic system for liver resection. In our study, however, there was no difference in total operative time between the two groups. Moreover, the total operative time tended to be shorter with RALR, thus it is unlikely that the robotic system influenced the longer parenchymal resection time. In a previous report, Lee et al. [30] concluded that the robotic system permits ALR with an extrahepatic Glissonean pedicle approach, particularly in cases involving the right liver, and can be safely performed in select patients. Kato et al. [31] also reported in detail on robotic liver resection focusing on the Laennec’s capsule and showed that robotic liver resection is a potentially feasible, safe, and acceptable oncologic platform of liver resection that is applicable to various types of hepatectomies. The main limitations of the present study were related to the limited number of patients and the retrospective nature of the analysis. Further prospective randomized controlled trials are necessary to affirm whether robotic surgery is likely to become the new standard procedure for liver resections.

In conclusion, extrahepatic GPI focusing on the Laennec’s capsule is the most reasonable method of inflow control because of procedure precision and safety. During laparoscopic surgery, however, the isolation of first- and second-order branches of the Glissonean pedicle with preservation of the Laennec’s capsule on the liver parenchyma is not difficult, but isolation of the third-order branches of the Glissonean pedicle is extremely difficult. This difficulty may be one of the reasons that widespread use of laparoscopic extrahepatic GPI has not been more widely adopted. Extrahepatic GPI is not indicated for all ALRs. For example, an intrahepatic GPI is safer and more common procedure than an extrahepatic GPI in the cone unit resection [5]. In this case, intraoperative ultrasound (IOUS) is mainly used to find anatomical landmarks in the liver and identify the responsible Glissonean pedicle based on their relationships. IOUS has long been one of the very useful instruments to guide anatomical liver resection [32]. Surgeons need to be proficient with IOUS on a regular basis because the quality of the operation depends on the quality of the IOUS, especially in the case of the laparoscopic surgery, which has small tactile sensation and severe motion restrictions. Thus, it is essential to use the two methods in different cases. Furthermore, in cases of cirrhosis liver, preservation of the Laennec’s capsule is relatively easy because it is slightly thickened. In cases of fatty liver or after chemotherapy, the parenchyma and membrane are very fragile and complete preservation of the Laennec’s capsule is difficult, so GPI from the hilum with parenchymal crushing is the preferred approach. We conclude, however, that the robotic system has the potential to make preservation of the Laennec’s capsule on the liver parenchyma easier and reduce the difficulty of extrahepatic isolation of peripheral branches of the Glissonean pedicle, and could significantly contribute to the generalization and widespread use of this technique.

## Figures and Tables

**Figure 1 jpm-13-01154-f001:**
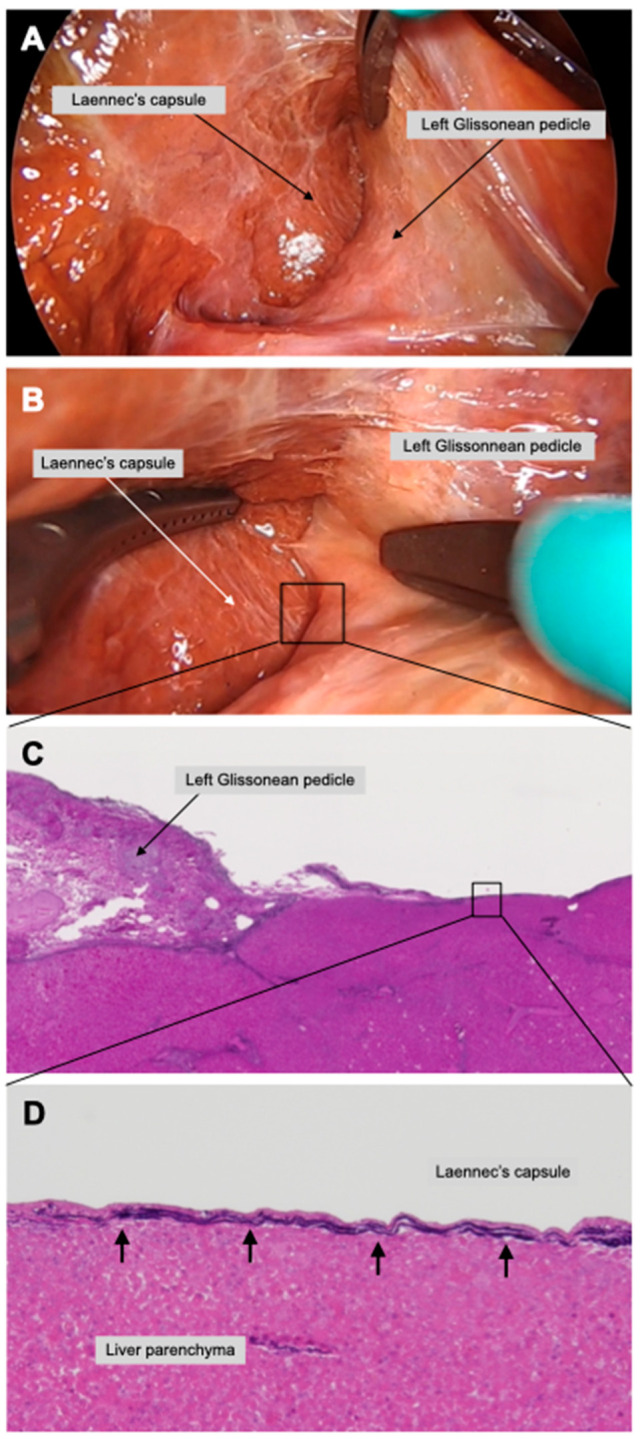
**Extrahepatic left Glissonean pedicle isolation in a cadaveric model.** After dissection between the arantius plate and the liver parenchyma, the dorsal aspect of the left Glissonean pedicle is non-blindly dissected while preserving the Laennec’s capsule (**A**). The right side of the left Glissonean pedicle and the Laennec’s capsule were separated (**B**). Pathologic examination using Elastica van Gieson staining proved the presence of the Laennec’s capsule in low- (**C**) and in high-power fields (black arrows)) (**D**).

**Figure 2 jpm-13-01154-f002:**
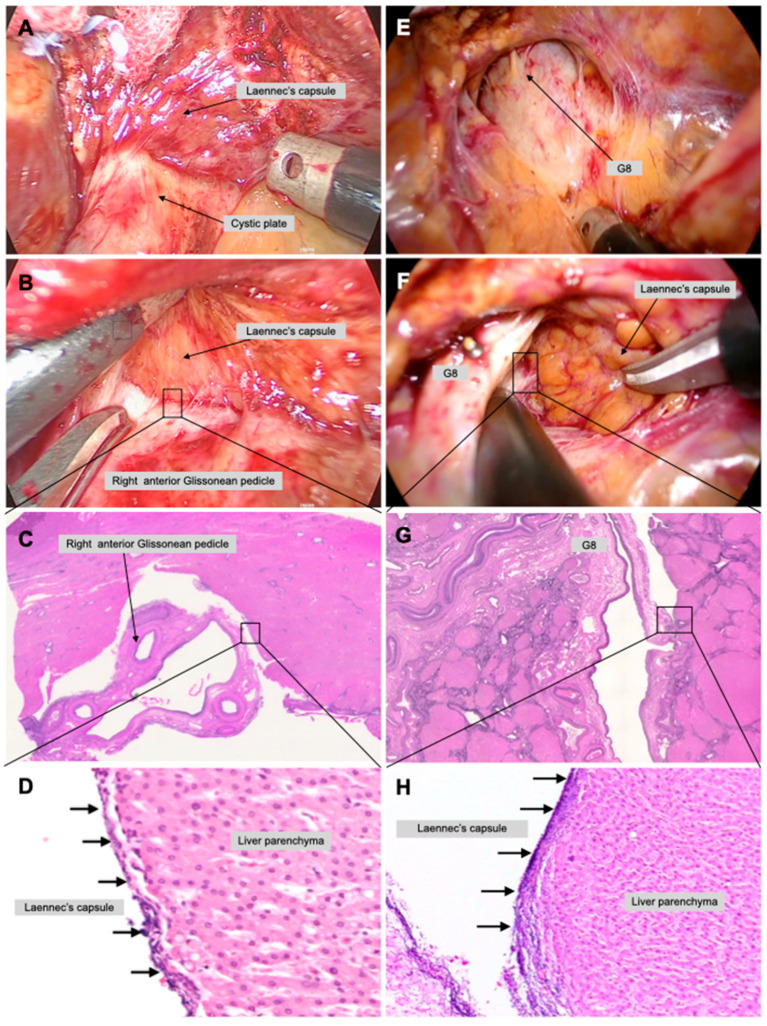
**Laparoscopic second and third branches of the Glissonean pedicle isolation in a live body.** Right anterior Glissonean pedicle isolation was performed in laparoscopic right hepatectomy to validate the appropriate layer to be dissected. The space between the cystic plate and the Laennec’s capsule was dissected to expose the Laennec’s capsule (**A**). The dissection in the same layer as the cystic plate cholecystectomy was continued to the ventral side of the right anterior Glissonean pedicle (**B**). The Laennec’s capsule on the liver parenchyma and exfoliated right anterior Glissonean pedicle of a live body is shown in low- (**C**) and high-power fields (black arrows) (**D**). Glissonean pedicle 8 isolation was performed during a laparoscopic segmentectomy 8. Blunt dissection between the right anterior Glissonean pedicle and the Laennec’s capsule in the direction of the peripheral Glissonean pedicle allowed identification of the Glissonean pedicle 8 (G8) and Glissonean pedicle 5 (G5) (**E**). The dissection was performed while looking directly at the dorsal aspect of G8 using the laparoscopic magnification effect (**F**). The Laennec’s capsule on the liver parenchyma and exfoliated G8 is shown in low- (**G**) and high-power fields (black arrows) (**H**).

**Figure 3 jpm-13-01154-f003:**
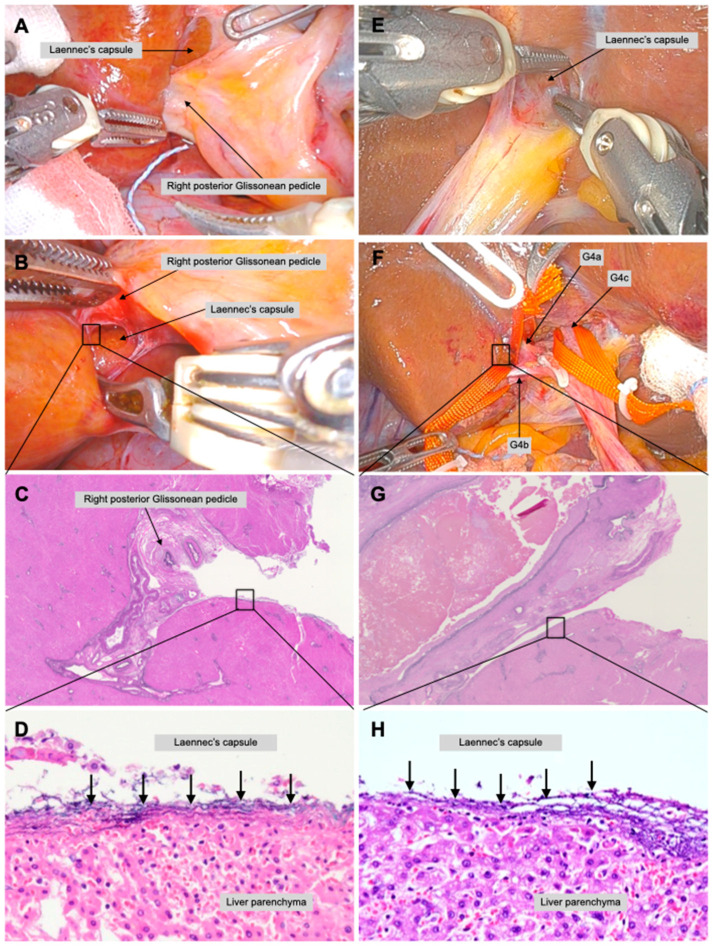
**Robotic second and third branches of Glissonean pedicle isolation in a live body.** Right posterior Glissonean pedicle isolation was performed during robotic right posterior sectionectomy. The junction of the right anterior and posterior Glissonean pedicles was identified and exposed the Laennec’s capsule (**A**). Blunt dissection between the Laennec’s capsule and the right posterior Glissonean pedicle from the caudal view while directly observing the dorsal view with a robotic endoscope (**B**). The Laennec’s capsule on the liver parenchyma and the exfoliated right posterior Glissonean pedicle of a live body is shown in low- (**C**) and high-power fields (black arrows) (**D**). Glissonean pedicle 4a (G4a), 4b (G4b), and 4c (G4c) isolation was performed in robotic segmentectomy 4b. The gap between the umbilical plate and the Laennec’s capsule was separated (**E**). All responsible branches of the Glissonean pedicle can be isolated without misidentification while directly observing the branches (**F**). The Laennec’s capsule on the liver parenchyma and the exfoliated Glissonean pedicle is in low- (**G**) and high-power fields (black arrows0 (**H**).

**Table 1 jpm-13-01154-t001:** Patient characteristics.

	Lap (n = 60)	Robot (n = 39)	
Age; mean ± SD (years)	69.3 ± 10.5	69.6 ± 9.5	N. S
Gender (male/female)	34/26	23/16	N. S
BMI; mean ± SD (kg/m^2^)	23.4 ± 3.9	22.6 ± 3.5	N. S
Liver cirrhosis; n (%)	15 (25)	9 (23)	N. S
ICG R15, %	10.1 ± 4.6	12.3 ± 5.7	N. S
Indication for liver resection, n (%)			
Hepatocellular carcinoma	29	18	N. S
Intrahepatic cholangiocarcinoma	6	4	
Symptomatic cyst	2	0	
Hemangioma	1	0	
Colorectal metastasis to liver	22	17	
Previous liver resection	8 (13%)	5 (13%)	
Open surgery; n (%)	3	3	N. S
Laparoscopic surgery; n (%)	4	1	
Robotic surgery; n (%)	1	1	

N. S: Not Significant.

**Table 2 jpm-13-01154-t002:** Types of liver resection.

	Lap (n = 60)	Robot (n = 39)	
Left lateral sectionectomy	5	5	N. S
Segmentectomy			
Segmentectomy 1	3	2	N. S
Segmentectomy 2	3	3	
Segmentectomy 3	1	3	
Segmentectomy 4a	1	1	
Segmentectomy 4b	2	4	
Segmentectomy 5	2	5	
Segmentectomy 6	4	3	
Segmentectomy 7	7	0	
Segmentectomy 8	5	3	
Left medial sectionectomy	1	2	N. S
Right anterior sectionnectomy	2	0	N. S
Right posterior sectionectomy	4	1	N. S
Left hepatectomy	12	5	N. S
Right hepatectomy	8	2	N. S

N. S: Not Significant.

**Table 3 jpm-13-01154-t003:** Peri- and post-operative outcomes.

	Lap (n = 60)	Robot (n = 39)	
Operative time; mean ± SD (min)	399.8 ± 109.5	366.1 ± 98.3	N. S
Duration of Glissonean pedicle isolation; mean ± SD (min)	32.9 ± 11.6	27.2 ± 9.3	*p* < 0.05
Measured blood loss; mean ± SD (mL)	289.7 ± 303.8	131.6 ± 102.2	*p* < 0.05
Iwate difficulty score; mean ± SD	7.7 ± 1.9	7.4 ± 2.1	N. S
Difficulty level; n (%)			
Intermediate	20 (33%)	12 (31%)	N. S
Advanced	32 (53%)	20 (51%)	N. S
Expert	8 (13%)	7 (18%)	N. S
Morbidity Clavien-Dindo ≥grade Ⅲa; n (%)	2 (3.3%)	2 (5.1%)	N. S
Bile leakage; n (%)	0	1	
Abdominal abscess; n (%)	2	1	
Mortality; n (%)	0	0	
Postoperative hospital stay; mean ± SD (day)	12.8 ± 8.5	11.3 ± 3.9	N. S

N. S: Not Significant.

## Data Availability

Data is unavailable due to privacy or ethical restriction.

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
