# Peer review of "Glissonean Pedicle Isolation Focusing on the Laennec’s Capsule for Minimally Invasive Anatomical Liver Resection"

_jpm, 2023, doi:10.3390/jpm13071154_

Round 1

Reviewer 1 Report

This is a well-written original paper on a hot topic in liver surgery.

I would just ask the authors to include comments about

- the possible intraoperative incidents using this approach

- the recorded major complications may be linked to this approach?

- the adverse factors that prevent in some cases the extraglissonian approach;

- does the background liver (cirrhosis, steatosis, etc) influence the feasibility of such approach?

Author Response

Thank you so much for your very thought-provoking points. We will respond to your comments upon careful review.

Reviewer 1: This is a well-written original paper on a hot topic in liver surgery. I would just ask the authors to include comments about

- the possible intraoperative incidents using this approach

Response 1: Thank you for your suggestion, I added the sentence below.

When performing the extrahepatic GPI, without a clear understanding of the layer structure around the Glissonean pedicle, the recognition of the boundaries is unclear and may enter into the Glissonean sheath and cause damage to the artery, portal vein, and bile ducts. (Line 14-17, P9)

- the recorded major complications may be linked to this approach?

Response 2: Thank you for your suggestion, I added the sentence below.

Complications in LALR were intra-abdominal abscess without bile leak in two cases; complications in RALR were bile leak and subcutaneous abscess without bile leak. The bile leak in RALR was unlikely to have been caused by extrahepatic GPI because the bile leak was from the peripheral bile duct and there was no injury to the hilar bile duct. (Line 22-25, P10)

- the adverse factors that prevent in some cases the extraglissonian approach;

Response 3: Thank you for your suggestion, I added the sentence below.

The extrahepatic GPI is not indicated for all ALRs. For example, cone unit resection [32] and 3rd order branches that bifurcate very peripherally may not be reasonable to perform the extrahepatic GPI, so the intrahepatic GPI is preferred. In this case, Intraoperative ultrasound (IOUS) is mainly used to find anatomical landmarks in the liver and identify the responsible Glissonean pedicle based on their relationships. IOUS has long been one of the very useful instruments to guide anatomic liver resection [33]. Surgeons need to be proficient with IOUS on a regular basis because the quality of the operation depends on the quality of the IOUS, especially in the case of laparoscopic surgery, which has small tactile sensation and severe motion restrictions. Thus, it is essential to use the two methods in different cases. (Line 8-16, P11)

- does the background liver (cirrhosis, steatosis, etc) influence the feasibility of such approach?

Response 4: Thank you for your suggestion, I added the sentence below.

In cases of cirrhosis liver, preservation of the Laennec’s capsule is relatively easy because that is slightly thickened. In cases of fatty liver or after chemotherapy, the parenchyma and membrane are very fragile and complete preservation of the Laennec’s capsule is difficult, so GPI from the hilum with parenchymal crushing is the preferred approach. (Line 16-19, P11)

Reviewer 2 Report

I have read with pleasure the paper by Morimoto et al. They propose a valid comparison between laparoscopic and robotic surgery in anatomic liver resection. However, there's no mention of intraoperative ultrasound, that nowadays is a fundamental tool in the identification of glissonian pedicles, much easier of extraepatic approach. Please mention the utilization of IOUS, both in laparoscopic and robotic surgery. 

English is fine.

Author Response

Thank you so much for your very thought-provoking points.
We will respond to your comments upon careful review.

Reviewer 2: I have read with pleasure the paper by Morimoto et al. They propose a valid comparison between laparoscopic and robotic surgery in anatomic liver resection. However, there's no mention of intraoperative ultrasound, that nowadays is a fundamental tool in the identification of glissonian pedicles, much easier of extraepatic approach. Please mention the utilization of IOUS, both in laparoscopic and robotic surgery. 

Response 1: Thank you for your suggestion.

Of course, the IOUS is one of the most important instruments in hepatic resection. However, in the present paper, we focus on the fact that the most accurate method of inflow control is the extrahepatic Glissonean approach. the IOUS is used only for tumor identification and for the intrahepatic Glissonean approach. The usefulness of IOUS is not what we should emphasize in our paper. We have added a description of the usefulness of IOUS in general.

The extrahepatic GPI is not indicated for all ALRs. For example, cone unit resection [32] and 3rd order branches that bifurcate very peripherally may not be reasonable to perform the extrahepatic GPI, so the intrahepatic GPI is preferred. In this case, Intraoperative ultrasound (IOUS) is mainly used to find anatomical landmarks in the liver and identify the responsible Glissonean pedicle based on their relationships. IOUS has long been one of the very useful instruments to guide anatomic liver resection [33]. Surgeons need to be proficient with IOUS on a regular basis because the quality of the operation depends on the quality of the IOUS, especially in the case of laparoscopic surgery, which has small tactile sensations and severe motion restrictions. Thus, it is essential to use the two methods in different cases. (Line 8-16, P11)

Round 2

Reviewer 2 Report

Thanks to the authors for the introduction of the suggested paragraph.